# Inheritance and Molecular Characterization of a Novel Mutated *AHAS* Gene Responsible for the Resistance of AHAS-Inhibiting Herbicides in Rapeseed (*Brassica napus* L.)

**DOI:** 10.3390/ijms21041345

**Published:** 2020-02-17

**Authors:** Qianxin Huang, Jinyang Lv, Yanyan Sun, Hongmei Wang, Yuan Guo, Gaoping Qu, Shengwu Hu

**Affiliations:** 1State Key Laboratory of Crop Stress Biology in Arid Areas, Yangling 712100, China; qianxinh@163.com (Q.H.); lvjy890727@163.com (J.L.); 15929800833@163.com (Y.S.); 18729362103@163.com (H.W.); guoyuan2109@163.com (Y.G.); qugp68@163.com (G.Q.); 2College of Agronomy, Northwest A&F University, Yangling 712100, China; 3College of Life Sciences, Northwest A&F University, Yangling 712100, China

**Keywords:** rapeseed (*Brassica napus* L.), acetohydroxyacid synthase, single-point mutation, herbicide resistance

## Abstract

The use of herbicides is an effective and economic way to control weeds, but their availability for rapeseed is limited due to the shortage of herbicide-resistant cultivars in China. The single-point mutation in the *acetohydroxyacid*
*synthase* (*AHAS*) gene can lead to AHAS-inhibiting herbicide resistance. In this study, the inheritance and molecular characterization of the tribenuron-methyl (TBM)-resistant rapeseed (*Brassica napus* L.) mutant, *K5*, are performed. Results indicated that TBM-resistance of *K5* was controlled by one dominant allele at a single nuclear gene locus. The novel substitution of cytosine with thymine at position 544 in *BnAHAS1* was identified in *K5*, leading to the alteration of proline with serine at position 182 in BnAHAS1. The TBM-resistance of *K5* was approximately 100 times that of its wild-type *ZS9*, and *K5* also showed cross-resistance to bensufuron-methyl and monosulfuron-ester sodium. The *BnAHAS1^544T^* transgenic *Arabidopsis* exhibited higher TBM-resistance than that of its wild-type, which confirmed that *BnAHAS1^544T^* was responsible for the herbicide resistance of *K5*. Simultaneously, an allele-specific marker was developed to quickly distinguish the heterozygous and homozygous mutated alleles *BnAHAS1^544T^*. In addition, a method for the fast screening of TBM-resistant plants at the cotyledon stage was developed. Our research identified and molecularly characterized one novel mutative *AHAS* allele in *B. napus* and laid a foundation for developing herbicide-resistant rapeseed cultivars.

## 1. Introduction

Rapeseed (*Brassica napus* L.), an important oilseed crop worldwide, is planted in approximately 75 million ha of land in China annually over the last decade [1] and supplies 60% of the vegetable oils in the Chinese market. In recent years, weed control is becoming a challenge in rapeseed fields due to the rapid extension of rapeseed mechanized production [2,3]. The use of herbicide-resistant varieties is an economical way to control weeds in modern agriculture [4]. However, the availability of selective herbicides for rapeseed is limited due to a shortage of herbicide-resistant cultivars in China [5,6]. To date, the primary measurement to control weeds in rapeseed fields in China is to apply acetochlor as a preemergence herbicide. In addition, two postemergence herbicides, chloropyridine acid and benazolin-ethyl, can be used for controlling broadleaf weeds in rapeseed fields [7].

More than one-sixth (54/302) of the globally registered herbicides are acetohydroxyacid synthase (AHAS, EC 2.2.1.6)-inhibiting herbicides [8]. These kinds of herbicides kill susceptible plants by suppressing the AHAS enzyme activity, also known as acetolactate synthase (ALS), which plays an important role in the biosynthetic pathway of branched-chain amino acids valine, leucine and isoleucine at the first step [9,10,11]. Since the first introduction of AHAS-inhibiting herbicides into the agronomic production in the 1980s, they have become a valuable tool in controlling weeds due to their low dosage, environmental friendliness, low mammalian toxicity, wide crop selectivity and high efficacy [8]. These kinds of herbicides can be classified into five groups, namely, sulfonylureas (SU), sulfonylamino-carbonyltriazolinones, imidazolinones (IMI), triazolopyrimidines and pyrimidinylthiobenzoates [12,13,14,15]. Tribenuron-methyl (TBM), an SU herbicide, was produced by the Dupont Company in the early 1980s and introduced to China in 1988 [16]. At present, TBM is used in broadleaf weed control across wheat fields in China and accounts for over half of the total herbicide usage due to its high efficacy at low dosage, low effect on nontarget organisms and high selectivity [17]. However, conventional rapeseed varieties are sensitive to TBM due to the lack of resistant genes. If the rapeseed variety with TBM-resistance can be developed, then the prescription of the combination of TBM with the existing monocotyledonous herbicides which are used in rapeseed fields can provide an alternative way to effectively control weeds in rapeseed fields. 

*AHAS* is particularly vulnerable to gene mutations and substitutions, which can convert *AHAS* from the herbicide-sensitive form to the herbicide-resistant form [9]. To date, the point mutations of *AHASs* primarily occur in their conservative domains. These mutations occur in the eight mutation sites Ala122, Pro197, Ala205, Asp376, Arg377, Trp574, Ser653 and Gly654 in AHASs (in reference to *Arabidopsis thaliana* L.) [18,19]. Three functional genes *BnAHAS1*, *BnAHAS2* and *BnAHAS3* and two pseudogenes *BnAHAS4* and *BnAHAS5* have been identified in rapeseed (*B. napus*) [20]. In *B. napus*, several AHAS-inhibiting herbicide-resistant mutants have been obtained through spontaneous and chemical mutagenesis by using ethyl nitrosourea or ethyl methyl sulfonate (EMS). These mutants include PM1 and M9 that harbor Asp instead of Ser at position 653 of BnAHAS1 (S653A in BnAHAS1), PM2 and M342 that harbor T574L in BnAHAS3 and M45 that harbors P197S/L in BnAHAS3 [6,21,22,23,24] and have been reported to be resistant to one or more dissimilar AHAS-targeted herbicides. These AHAS mutants have been used to develop herbicide-resistant rapeseed varieties [25] and are recommended to be used as male parents in chemical-induced male sterility hybrid seed production [21,26]. 

Herbicide-resistant rapeseed was first registered and cultivated in the 1980s in Canada. At present, the herbicide-resistant rapeseed varieties are dominant in the seed market of Canada [27]. The primary types of herbicide-resistant rapeseed in Canada include the glufosinate-, glyphosate- and IMI-resistant varieties registered by the Bayer, Monsanto and BASF companies, respectively. These herbicide-resistant rapeseed varieties are subject to international intellectual property protection, in which expensive patent fees should be paid for their commercial cultivation. In addition, the glyphosate- and glufosinate-resistant rapeseed are transgenic varieties, which are not yet approved for commercial cultivation in China. At present, China lacks herbicide-resistant cultivars with independent intellectual property, which has become the bottleneck of the rapeseed industry. Therefore, the development of herbicide-resistant rapeseed varieties and their corresponding herbicide formulations are important for the development of the rapeseed industry.

Previously, our research group has identified three TBM-resistant rapeseed mutants derived from *Zhongshuang No.9* (*ZS9*) via EMS mutagenesis and TBM foliar-spray screening [5,28]. In this study, one of the TBM-resistant rapeseed mutants (*K5*) was characterized. The objectives of the present investigation were to: (1) reveal the resistance or cross-resistance of *K5* to different herbicides, (2) determine the mode of inheritance and the molecular mechanisms of herbicide resistance and (3) develop a rapid method for screening herbicide-resistant materials. The obtained results will lay a foundation for developing herbicide-resistant varieties in rapeseed.

## 2. Results

### 2.1. Cross-Resistance of the Mutant Line K5 to Different Herbicides

*K5* and *ZS9* were treated with 10 different herbicides at the 4–6 leaf stage to investigate the responses of the mutant line *K5* to different herbicides. The tested herbicides and the used rates are detailed in the Section 4.1. Results indicated that the line *K5* exhibited resistance to three herbicides: TBM, bensufuron-methyl (BSM) and monosulfuron-ester sodium (MES) (Appendix A). However, it was susceptible to imidazole nicotinic, florasulam (FU), carfentrazone-ethyl (CFE), sulfometuron methyl ester (SME), nicosulfuron (NSF), glyphosate and glufosinate\. The line *ZS9* was almost susceptible to all the application rates of the tested herbicides. 21 DAT (days after treatment), the *ZS9* individual plants showed a variety of symptoms, such as yellow, curly and withered leaves and purple veins, which were particularly evident in newly growing leaves (Appendix A). 

Five traits, namely, phytotoxicity index, the leaf angle, leaf numbers, fresh weight and dry weight, were investigated after treating the lines *ZS9* and *K5* with three herbicides (TBM, BSM and MES) at different rates. The analysis of variance indicated that only BSM treatments have no effect on leaf angle, leaf numbers, fresh weight and dry weight of *K5*, except traits of lines *ZS9* and *K5* under other tested rates of herbicides were remarkably affected (Appendix A).

The phytotoxicity index and the inhibition rates of the leaf angle, leaf numbers and fresh and dry weights of *K5* increased more slowly compared with those of *ZS9* at increasing TBM rates (Appendix A). *ZS9* showed low-level resistance to TBM. Under the treatment of 0.06 g a.i. ha^−1^ TBM in a volume of 300 L ha^−1^, no apparent phytotoxicity was observed in *ZS9* individuals 21 DAT (Appendix A). However, *ZS9* showed a variety of phytotoxicity when treated with other rates of TBM ≥ 0.15 g a.i. ha^−1^, and all individual plants died under the treatment of 0.6 g a.i. ha^−1^ TBM (Figure 1 and Table 1). The mutant line *K5* showed high-level resistance to TBM. No apparent difference was observed between the TBM-free treatment and TBM treatments at rates up to 15 g a.i. ha^−1^ (Appendix A). When the TBM rate increased to 30 g a.i. ha^−1^, only young leaves were injured. When the TBM rate increased to 60 g a.i. ha^−1^, individual plants were injured (Appendix A); however, most of the plants can survive after winter. The fresh and dry weights *I*_50_ of *K5* were approximately 70 and 97 times, respectively, that of *ZS9* (Table 1). The lethal rate of TBM for *K5* was at least 100 times that of *ZS9* (Table 1). The safety rate of TBM for *K5* was 15 g a.i. ha^−1^, which was approximately 250 times that of *ZS9* (Table 1).

The experimental results of BSM indicated that significant differences existed among the tested rates of BSM for the five traits of *ZS9.* However, except for the phytotoxicity index, the traits (leaf angle, leaf numbers, and fresh and dry weights) did not show significant differences for *K5* (Appendix A). *ZS9* was almost normal after being treated with different rates of BSM ≤ 0.15 g a.i. ha^−1^. *ZS9* individuals were slightly injured after being treated with 0.3 g a.i. ha^−1^ BSM (Appendix A). The highest rate of BSM used was not lethal for *K5* (Appendix A). *K5* appeared almost normal after being treated with different rates of BSM ≤ 18 g a.i. ha^−1^. Based on the fresh and dry weights *I*_50_, *K5* showed approximately 55 and 30 times BSM resistance, respectively, that of *ZS9* (Table 1). The lethal and safety rates of BSM for *K5* were at least 18 and approximately 120 times, respectively, that of *ZS9* (Table 1). Therefore, *K5* conferred 30 times BSM-resistance than that of the *ZS9*.

The MES results indicated that significant differences existed among the tested rates of MES in the five traits of the *ZS9* and *K5* lines (Appendix A). *ZS9* was almost normal after being treated with different rates of MES ≤ 0.1 g a.i. ha^−1^. *ZS9* individuals were slightly injured after being treated with 0.2 g a.i. ha^−1^ MES (Appendix A). *K5* appeared almost normal after being treated with different rates of MES ≤ 10 g a.i. ha^−1^. Based on the fresh and dry weights *I*_50_, *K5* showed approximately 49 and 55 times MES resistance, respectively, that of *ZS9* (Table 1). The lethal and safety rates of MES for *K5* was approximately 30 times and 100 times, respectively, that of *ZS9* (Table 1). Therefore, *K5* conferred 30 times MES resistance than that of *ZS9*.

In general, the mutant line *K5* conferred a high-level resistance to TBM and showed resistance to BSM and MES. The safety rate of TBM for *K5* (15 g a.i. ha^−1^) approached the recommended dosage for weed control in wheat crops (13.5–22 g a.i. ha^−1^).

### 2.2. AHAS Enzyme Activity Assay

The enzyme activities of *K5* and *ZS9* seedlings were tested after being treated with different rates of TBM, BSM and MES to determine whether the resistance to three herbicides (TBM, BSM and MES) in the mutant line *K5* is the result of the gain-of-function mutation of AHAS. The molecular weight of AHAS extracted from *ZS9* and *K5* seedlings free from any herbicide treatments was approximately 58–66 kDa (Figure 2d). The AHAS-specific activities in *ZS9* and *K5* seedlings were 0.098 ± 0.011 and 0.075 ± 0.013 U/mg, respectively. Significant differences did not exist between the two lines (t = 2.8, *p* = 0.13), which indicated that the mutation in *K5* did not affect its AHAS activity. The AHAS relative enzyme activity of the *ZS9* decreased suddenly with the increased TBM rate, particularly for the rates of TBM ≥ 0.6 g a.i. ha^−1^ (Figure 2a). By contrast, the AHAS relative enzyme activity of *K5* decreased gradually with the increased TBM rate, and TBM rates < 6 g a.i. ha^−1^ rarely affected the AHAS activity (Figure 2a). The AHAS activity of *K5* can be inhibited to 50% of its original level (*I*_50_) with 41.1 g a.i. ha^−1^ TBM treatment, which was approximately 250 times that of *ZS9* (*I*_50_ = 0.17 g a.i. ha^−1^). Under the treatment of BSM or MES, changes in the enzyme relative activity of *K5* and *ZS9* seedlings showed a similar trend to that of TBM treatments (Figure 2b,c). The in vitro enzyme activity of AHAS treated with different rates of TBM was also detected, and results indicated that the AHAS activity of the *ZS9* was suppressed by adding TBM solution and decreased by adding 1.0 mg L^−1^ TBM. However, the AHAS activity of *K5* was not inhibited significantly at the test rates of TBM (Appendix A). Simultaneously, the AHAS enzyme activity was tested from herbicide-sprayed materials or tested the in vitro way, but results revealed that the AHAS enzyme activity of *K5* was less inhibited than that of *ZS9* when treated with three different herbicides, which indicated that the resistance of *K5* was due to a gain-of-function mutation in the *AHAS* genes and not because of the overproduction of AHAS.

### 2.3. Genetic Investigation of TBM-Resistance of the Mutant Line K5

The seedlings of F_1_, BC_1_ and F_2_ progenies derived from *K5* and two susceptible parents, *SH11* and *ZS9*, were foliar sprayed with 4.5 g a.i. ha^−1^ TBM at the 4–6 leaf stage to determine the resistance inheritance of *K5*. Results indicated that two F_1_ populations from all reciprocal crosses were resistant to TBM (Table 2), and individual plants did not die in the whole growth period. The F_2_ population derived from the cross between *K5* and *ZS9* was segregated at a ratio of 1059 resistant plants:350 susceptible plants. The BC_1_ population was segregated at a ratio of 114 resistant plants:103 susceptible plants. The goodness-of-fit test indicated that the segregation of resistant and susceptible plants fits the expected Mendelian ratios of 3:1 and 1:1, respectively (χ^2^_c_ = 0.02, *p* = 0.89 and χ^2^_c_ = 0.28, *p* = 0.60; Table 2). The F_2_ and BC_1_ populations derived from the cross between the *K5* and the *SH11* also obtained similar results (χ^2^_c_ = 0.25, *p* = 0.62 and χ^2^_c_ = 0, *p* = 1; Table 2). These results indicated that the TBM resistance of *K5* was controlled by one pair of nuclear genes, which had dominant resistance but no cytoplasmic effect.

### 2.4. Sequence Comparison of the Mutant K5 and Wild-type (WT) BnAHAS Genes

Three functional *BnAHAS* genes (*BnAHAS1, BnAHAS2* and *BnAHAS3*) were amplified in *K5* and *ZS9* by using previously reported primers to determine the critical mutation(s) responsible for the herbicide resistance of the mutant line *K5* [29] (Appendix A). In addition, *BnAHAS1* was also amplified from five other susceptible lines, namely, *Zhong 7*, *Zhong 2*, *QSC*, *Q7C* and *S11R*. Results indicated that the *BnAHAS2* and the *BnAHAS3* sequences were identical between the lines *ZS9* and *K5*. Notably, a novel single nucleotide polymorphism (SNP) was detected at position 544 in the *BnAHAS1* coding region of *K5* with the alternation from C to T (Figure 3). The sequencing of *BnAHAS1* in five other lines (*Zhong 7*, *Zhong 2*, *QSC*, *Q7C* and *S11R*) did not detect this SNP. These lines showed C as the *ZS9* at position 544 in *BnAHAS1*. This mutant allele in the *K5* was designed as *BnAHAS1^544T^*, and its sequence was deposited in the GenBank (accession number: KP985786). The SNP at position 544 (C554T) led to a substitution of P with S at position 182 of BnAHAS1 (Figure 3), which corresponded to Pro-197 in AtAHAS. This finding suggested that this SNP was probably responsible for the herbicide-resistance of *K5*.

### 2.5. Development of Allele-Specific (AS) Markers and Co-Segregation Analysis

Screening homozygous herbicide-resistant individuals with the mutant allele *BnAHAS1^544T^* by spraying herbicides takes time, and using polymerase chain reaction (PCR)-based AS markers was expected to be a quick way to do such work. Therefore, several sets of AS–PCR primers were designed. One pair of AS–PCR markers (*BnA1*-F1 and *BnA1*-R4), which can distinguish homozygous-resistant individuals from heterozygous ones, was selected (Appendix A). Given that the *BnAHAS1* and *BnAHAS3* were highly homologous to each other in their coding regions, the forward primer *BnA1*-F1 was designed in the 5′ UTR region of *BnAHAS1* to avoid the amplification of *BnAHAS3*. The amplified products of the lines *K5* and *ZS9* were approximately 805 bp in length by the pair of primers *BnA1*-F1 and *BnA1*-R4. The lines *ZS9* and *K5* shared an *Ava*II restriction site in the amplified products of *BnAHAS1* at position 465. However, in the B*nAHAS1* sequence of *ZS9**,* another restriction site of *Ava*II was identified at position 544, where the SNP was located. This restriction site can be used to distinguish the WT allele of *BnAHAS1* from the mutant allele *BnAHAS1^544T^* in *K5*. As a result, the amplified product of *BnAHAS1* from *ZS9* can be digested into three fragments with the lengths of 644, 81 and 80 bp, respectively. However, the agarose gel electrophoresis can only detect two bands: 644 bp and about 80 bp in length. Two fragments with the lengths of 644 and 161 bp can be detected from the homozygous-resistant mutant (with allele *BnAHAS1^544T^*). The heterozygous-resistant individuals showed three bands with lengths of 644, 161 and 80 bp on the agarose gel. The individual plants of *ZS9* and *K5* with their corresponding F_1_ were detected with the AS–PCR marker, and results were perfectly obtained, as expected (Figure 4a). Moreover, the PCR amplification result of 30 susceptive *B. napus* accessions (Appendix A) with this pair of AS–PCR primers displayed the same band pattern as the susceptible parent *ZS9*, which validated the specificity of this marker (Figure 4b).

Co-segregation analysis was conducted using the AS–PCR marker in the BC_1_ population, which was derived from the cross between *K5* and *ZS9*, to figure out the relationship between herbicide resistance and the mutation (*BnAHAS1^544T^*) in *K5.* The BC_1_ population was separated into half-resistant and susceptive plants after spraying 4.5 g a.i. ha^−1^ TBM. The BC_1_ population was detected in the *BnAHAS1^544^* SNP_S_ by using the AS–PCR marker. The BC_1_ population of 273 individuals was evaluated, and results indicated that the AS-PCR markers co-segregated with the phenotyping results (Appendix A). 

### 2.6. Expression of BnAHAS1^544T^ Confers Herbicide Resistance in A. Thaliana Plants

A binary expression vector containing *BnAHAS1^544T^* was transformed into *WT Arabidopsis* to determine whether the mutant allele *BnAHAS1^544T^* conferred TBM resistance. Five transgenic lines (*TA1-1–TA1-5*) were obtained. After being selfed for two generations, two homozygous lines (*TA1-1* and *TA1-4*) were selected from the offspring and used for the following assays. First, these two homozygous lines were foliar sprayed with 0.3 g a.i. ha^−1^ TBM, and results indicated that these homozygous lines can tolerate this rate of TBM and did not show any symptoms, whereas *WT* plants died (Figure 5a). Second, the expression of *BnAHAS1* in *WT* and *TA1* was detected by semiquantitative PCR (SQ-PCR), and results revealed that the specific fragments of *BnAHAS1^544T^* can be amplified in the two transgenic lines. However, the fragment cannot be amplified in *WT*, which further verified that the mutated allele *BnAHAS1^544T^* contributed to the TBM resistance of *A. thaliana* (Figure 5b). Third, after spraying with different rates of TBM, the fresh weight of the transgenic line *TA1-1* decreased gradually, although the rates used were higher than those for WT (Figure 5c), which showed a similar trend as rapeseed lines *K5* and *ZS9* after treated with TBM (Appendix A). Collectively, *BnAHAS1^544T^* was concluded to endure the *K5* TBM resistance.

### 2.7. Development of a Method for Fast-screening TBM-Resistant Mutants in Greenhouse

Developing a fast way for screening TBM-resistant mutants in a greenhouse is important, because finding natural or mutated herbicide-resistant individuals in field is a time- and labor-consuming work. The seeds of *K5* and *ZS9* were sown in pots and immediately irrigated with five and seven different rates of TBM, respectively, to investigate the suitable TBM rate for distinguishing resistant individuals from sensitive ones (see the Section 4.9). After 5–7 days, all treated seeds started to emerge, with no apparent difference between *K5* and *ZS9*. After two weeks, TBM rates at 0.0003 and 0.003 g a.i. ha^−1^ did not inhibit the seedling growth of *ZS9*. However, TBM rates ≥ 0.03 g a.i. ha^−1^ inhibited seedling growth before the first leaf appeared (Figure 6). For *K5*, all the TBM rates ≤ 0.15 g a.i. ha^−1^ did not inhibit seedlings growth. Some seedlings were slightly inhibited at a 0.75 g a.i. ha^−1^ TBM rate, and seedling growth was inhibited at a 3.75 g a.i. ha^−1^ TBM rate (Figure 6). In general, TBM rates of 0.03–0.75 g a.i. ha^−1^ could distinguish a TBM-resistant line (*K5*) from a TBM-susceptible line (*ZS9*). In view of the relative safety and effectiveness, 0.15 g a.i. ha^−1^ TBM was utilized in further assays.

Two parents, *K5* and *ZS9*, and their derived F_1_, BC_1_ and F_2_ populations, were treated with 0.15 g a.i. ha^−1^ TBM to verify the feasibility of this fast-screening method on the basis of the described procedure above. Two weeks later, the seedlings of the F_1_ population grew normally. However, a phenotypic separation was observed in BC_1_ and F_2_ populations. Part of the seedlings grew normally, and the other part grew abnormally, as the herbicide-susceptible parent *ZS9* (Appendix A). The separation ratio of BC_1_ and F_2_ was analyzed (Appendix A), and the result was consistent with the genetic investigation of TBM-resistance based on field data (Table 2). In addition, 50 resistant and 50 susceptible individuals were collected from the BC_1_ population, and these individuals were genotyped using the AS–PCR marker developed previously. Phenotyping and genotyping perfectly matched each other.

As a result, a fast way to screen TBM-resistant individuals in a greenhouse was developed. The seeds of a candidate rapeseed population were sown in pots and immediately irrigated with 0.15 g a.i. ha^−1^ TBM. Two weeks later, TBM-resistant and TBM-susceptible individuals can be distinguished on the basis of phenotyping. 

## 3. Discussion

The use of herbicides is an effective and economic way to control weeds in rapeseed fields, and the successful control of weeds can avoid yield loss [27]. In China, the effective control of weeds has become an urgent challenge due to the rapid development of mechanization in rapeseed production. However, the availability of selective herbicides for rapeseed is quite limited due to a shortage of herbicide-resistant cultivars in China [5,6]. Finding new nontransgenic herbicide-resistant rapeseed materials is necessary due to the limitation of herbicide-resistant cultivars with independent intellectual property rights and the prohibition of commercial production of transgenic rapeseed in China. At present, many other crops are found harboring the AHAS-inhibiting herbicide resistance by nontransgenic ways due to the single-point mutation in *AHAS* genes [25,30,31]. In addition, several rapeseed mutants conferring the AHAS-inhibiting herbicide resistance have been reported and proven due to mutations in BnAHASs, such as BnAHAS1-S653A [6,24], BnAHAS3-S197L [21] and BnAHAS3-T574L [24,32]. In our laboratory, seeds of rapeseed *ZS9* (*B. napus*) are mutated through treatment with 1.0% (v/v) EMS solution, and three TBM-resistant individuals are selected by the foliar spraying of the TBM solution [5,28]. In the previous study, the mutant line *K5* is a novel valuable germplasm for hybrid breeding through chemical hybridization agents, and the hypothesis that AHAS should be the target of the AHAS-inhibiting herbicide TBM when used as a chemical hybridization agent in rapeseed is supported [26]. In the present study, the inheritance and molecular characterization of the mutant line *K5* was carried out. Genetic results indicated that the resistance in *K5* was controlled by a dominant allele at a single nuclear gene locus, without a cytoplasmic effect. Molecular analysis revealed that a novel substitution of cytosine with thymine at the position 544 in *BnAHAS1* was identified in the *K5* (this mutant allele in the *K5* was designed as *BnAHAS1^544T^*), leading to the alteration of Per with Ser at position 182 in BnAHAS1 (correspondence to *A. thaliana* AtAHAS-197). The safety rate of TBM for *K5* growth was approximately 15 g a.i. ha^−1^, whereas that of *ZS9* was lower than 0.15 g a.i. ha^−1^, indicating that the TBM resistance of *K5* was approximately 100 times that of *ZS9* (Table 1). This mutant allele *BnAHAS1^544T^* also conferred resistance to two other herbicides, BSM and MES. Collectively, the mutant *BnAHAS1^544T^* provided a novel invaluable resource for hybrid breeding by chemical hybridization agents and herbicide-resistance breeding in rapeseed. 

The amino acid residue 197 of the AHAS enzyme, which makes up the SU binding pocket, is located at the fifth-helix in the α domain of the AHAS catalytic subunit and involved in anchoring the aromatic ring through hydrophobic interactions [9,10]. In addition, the substitution of P197 in AHAS has been reported to result in herbicide resistance in *A. thaliana* [33], soybean [31] and many weeds [34,35]. In the present study, the transgenic *BnAHAS1^544T^ Arabidopsis* experiment revealed that the overexpression of the *BnAHAS1^544T^* allele conferred TBM resistance (Figure 5a). Furthermore, the AS–PCR markers, which were developed on the basis of the *BnAHAS1^544T^* allele, were also co-segregated with herbicide resistance. Collectively, our experiments confirmed that the newly found substitution at position 182 of BnAHAS1 (correspondence to *A. thaliana* AtAHAS-197) in *K5* provided herbicide resistance. 

Herbicide-resistance mechanisms can be generally classified into two categories: target-site resistance (TRS) and non-target-site resistance (NSTR) [36]. The TSR mechanism largely involves mutation(s) in the target site of the action of an herbicide, resulting in an insensitive or less sensitive target protein of the herbicide [19,36]. The NTSR is endowed by any mechanism not belonging to the TSR, which include reduced herbicide uptake/translocation, increased herbicide metabolism, decreased rate of herbicide activation and/or sequestration [37,38]. Our experimental results indicated the herbicide-resistance in rapeseed mutant line *K5* is caused by the alteration of Per with Ser at position 182 in BnAHAS1, resulting in the less sensitive target protein BnAHAS1 of the herbicides. Therefore, the herbicide-resistance of the mutant *K5* belongs to the TSR. Whether TSR and NTSR mechanisms coexist in the mutant *K5* is necessary to investigate in the future [37]. It is expected that the “omics” strategy (genomics, transcriptomics, proteomics or metabolomics) will play an important role to unravel the genetic base of NTSR in rapeseed [38].

The availability of suitable molecular markers facilitated the effective screening of herbicide-resistant accessions in the rapeseed breeding program. Hu et al. [6] developed the AS–PCR marker to distinguish IMI-resistant materials, which contained the BnAHAS1-S653A resistant allele. Li et al. [21] developed the derived cleaved amplified polymorphic sequences (CAPS) marker to differentiate TBM-resistant materials, which contained the BnAHAS3-S197L allele. In addition, Hu et al. [32] developed the CAPS markers to differentiate a TBM- and IMI-resistant line M342, which contained the BnAHAS3-T574L resistant allele. In the present study, one pair of AS–PCR primers was developed on the basis of the restriction sites of the endonuclease *Ava*II, which can effectively identify homozygous and heterozygous individuals with the resistant *BnAHAS1^544T^* allele (Figure 4a). The AS–PCR marker also co-segregated with the herbicide resistance in the BC_1_ population derived from the cross between *K5* and *ZS9* (Appendix A), which suggested that this marker can be used in future rapeseed breeding for herbicide resistance.

Combination of marker-aided selection and fast-screening of herbicide-resistant seedlings in greenhouses will speed up the herbicide-resistant breeding program. Therefore, a method for fast-screening TBM-resistant materials in greenhouses, which employed 0.15 g a.i. ha^−1^ TBM to irrigate pots that sow the candidate seed population of herbicide-resistance, was developed. After two weeks, the herbicide-resistance of the candidate population was assessed on the basis of the inhibition of seedling growth at the cotyledon stage (Figure 6). This method was applied to F_2_ and BC_1_ populations derived from the cross between *K5* and *ZS9*. The phenotyping results corresponded well with the genotyping results by using our AS–PCR marker (Appendix A). In addition, Magha et al. [39] employed a seed-soaking treatment to bioassay the herbicide-resistance of a rapeseed mutant and suggested that the method can be used to screen tolerant plants in inbreeding progenies or backcross-derived populations, as well as screen rapeseed mutants tolerant to herbicide in populations derived from seed mutagenesis. The seed-soaking treatment is successfully used in selecting soybean mutants tolerant to sulfonylureas [40,41]. Our method was similar to the seed-soaking method reported by Magha et al. [39]. The difference is that our method directly sowed seeds in pots and irrigated the pots with herbicide solution, which was slightly simpler than that reported by Magha et al. Simultaneously, a fast-screening method in the greenhouse was developed to select the herbicide-resistant individuals in the seedling stage for the herbicide breeding program.

## 4. Materials and Methods

### 4.1. Plant Materials and Herbicides 

A total of 33 rapeseed (*B. napus*) accessions, namely, *ZS9*, *K5*, *SH11* and 30 other rapeseed lines, were used in this investigation (Appendix A). *ZS9* was developed by the Oil Crops Research Institute of the Chinese Academy of Agricultural Sciences (Wuhan, China) and selfed for more than eight generations before use in the present study. *K5* is a TBM-resistant mutant line derived from *ZS9* via the EMS mutagenesis and obtained by TBM foliar spraying screening [5,28]. *SH11* is an elite restorer line. The 30 rapeseed lines (Appendix A) were used for AHAS sequence conformation and AS marker verification. By the end of September, all rapeseed materials were sown in the experimental field of Northwest A&F University (N 34.29°, E 108.06°), Yangling, Shaanxi, China, in 2014–2017. Rapeseed materials were grown in 2.0-m-long rows with 0.5 and 0.1-m-spacing between and within rows, respectively. Before sowing, 600 kg ha^−1^ of N, P and K compound fertilizer (N:P:K = 15:15:15) and 75 kg ha^−1^ urea with a nitrogen content of 46.7% were applied as a basal fertilizer. At the end of December, rapeseed seedlings were watered for their overwintering. Rapeseed seedlings were sprayed accordingly with different herbicides at the 4–6 leaf stage. The resistance was evaluated 21 DAT.

*A**. thaliana* (Col-0 ecotype) plants and its transgenic variants were grown at 22 °C under a 16-h light/8-h dark cycle (light intensity 70–150^–2^ s^–1^) and approximately 60% relative humidity in a phytotron.

The 10 different herbicides and their rates used in the present study are shown in Table 3. TBM (MaiFa^®^, 10% active ingredients); BSM (DaoWucao^®^, 10% active ingredients) and FU (FuMeiShi^®^, 50 g L^−1^ active ingredients) were produced by Hetian Chemical Co. Ltd. (Shanyang, China); Kuaida Agrochemical Co. Ltd. (Jiangsu, China) and Agricultural Hormone Engineering Hormone Co. Ltd. (Jiangsu, China), respectively. MES (97.5% active ingredients) was kindly provided by Professor Zhengming Li of NanKai University, Tianjin, China. Imidazole nicotinic (BaiLongTong^®^, 240 g L^−1^ active ingredients); CFE (FuShiMei^®^, 240 g L^−1^ active ingredients); SME (LvZhong^®^, 75% active ingredients); NSF (WuJingXiang^®^, 40 g L^−1^ active ingredients); glyphosate (ChuXinjie^®^, 80% active ingredients) and glufosinate (NuoDun^®^, 200 g L^−1^ active ingredients) were produced by LongDeng Chemical Co. Ltd. (Jiangsu, China); Plant Protectant Co. Ltd. (Suzhou, China); HuiFeng Agrochemical Co. Ltd. (Jiangsu, China); Plant Protection Institute of Chinese Academy of Agricultural Sciences Pharmaceutical Factory (Langfang, China); YunFan Chemical Co. Ltd. (JiangSu, China) and GuiHe Biotechnology Co. Ltd. (ShanDong, China), respectively.

### 4.2. Cross-Resistance of the Mutant line K5 to Different Herbicides

Ten different herbicides were used to treat the seedlings of rapeseed lines *K5* and *ZS9* at the 4–6 leaf stage at different rates (Table 3). 21 DAT, the symptoms were recorded in accordance with the method described by Gao et al. [42]_ENREF_32. The following indices were scored for each herbicide treatment. Phytotoxicity was scored at seven grading standards: 0, all leaves are green; 1, young leaves (the first and second ones) are light yellow–green; 2, part of young leaves are yellow; 3, the second leaf is yellow and curly; 4, the mature leaves are yellow–green or light purple; 5, certain mature leaves are dead and 6, the plant is dead. The phytotoxicity index was calculated using the following formula: (1)$$\mathrm{Phytotoxicity}\;\mathrm{index}=\sum \frac{\mathrm{score}\;\mathrm{of}\;\mathrm{the}\;\mathrm{standard}\;\times \;\mathrm{No.}\;\mathrm{of}\;\mathrm{plants}\;\mathrm{for}\;\mathrm{correspongding}\;\mathrm{standard}}{\mathrm{total}\;\mathrm{No.}\;\mathrm{of}\;\mathrm{plants}\;\times 7}$$

The leaf angle, leaf numbers and fresh and dry weights were investigated in accordance with the method described by Xin et al. [43]. 

The experiments were conducted in 2017. The field trials were arranged in a completely random block design with three replications for each line. Each plot contained four 2.0-m-length rows, with 0.5 and 0.1-m-spacing between and within rows, respectively. All traits were recorded from 10 seedlings for each replication, and the trait value was represented by the average data of 10 seedlings. 

### 4.3. Enzyme Extraction and Assays of AHAS Activity

In 2017, the rapeseed seedlings of the lines *ZS9* and *K5* at the 4–6 leaf stage were sprayed with three different herbicides (TBM, BSM and MES) at different rates (Table 3). The leaves from 10 seedlings of each herbicide rate treatment of both lines were collected 21 DAT, and the AHAS enzyme activity was assayed on the basis of the protocol of Lv et al. [26]. The specific activities of AHAS of both lines were estimated by using the zero-herbicide control. The AHAS enzyme extracted from the controls of both lines was also used for the in vitro assay of the AHAS enzyme in accordance with the protocol of Li et al. [21]. The concentrations of TBM used in the in vitro assay were 0, 1, 2, 5, 10 and 20 mg L^−1^. Three biological replications were included for each assay. The data from each line were fitted to a nonlinear regression model by using the curve fit of the SigmaPlot 12.0 software (Systat Software, San Jose, CA, USA). The nonlinear regression was based on a logistic function that was previously described mathematically [44]:(2)$$y=C-\left(D-C\right)/{\left[1+x/I_{50}\right]}^{b}$$
where *y* is the AHAS activity (% of the mean of the zero herbicide control), *x* is the concentration of TBM used in the enzyme assay, *C* and *D* are the lower and upper asymptotes, respectively, of the AHAS activity, *I*_50_ is the herbicide dose required to reduce the AHAS activity by 50% and *b* is the slope of the curve at approximately *I*_50_. The means of each treatment were estimated using the PROC ANOVA of the IBM SPSS Statistics 20.0 [45] and plotted on the logistic dose–response curves.

### 4.4. Genetic Analysis of TBM-Resistance in the Mutant K5

The mutant line *K5* was crossed with the lines *ZS9* and *SH11* to develop F_1_. F_1_ plants were self-pollinated and backcrossed to the susceptible parents to obtain the F_2_ and BC_1_ generations, respectively. By the end of September of the two crop seasons of 2016–2017, the seeds of all the materials, including parents, F_1_, F_2_ and BC_1_ progenies, were sown in the experimental field of Northwest A&F University. At the 4–6 leaf stage, the seedlings of the abovementioned rapeseed materials were sprayed with 4.5 g a.i. ha^−1^ TBM in a volume of 300 L ha^−1^. The resistance of the parents and their progenies was evaluated 21 DAT. Plants were scored as resistant when no herbicide damage or only slight injury was observed and sensitive when the plants died. The segregation of each population was assessed using the chi-square (χ^2^_c_) goodness-of-fit test.

### 4.5. Amplification and Sequence Analysis of BnAHASs 

The genomic DNA of seven rapeseed lines, namely, *ZS9*, *K5*, *Zhong 7*, *Zhong 2*, *QSC*, *Q7C* and *S11R* (Appendix A), were extracted from 0.5 g young leaves by using the cetyltrimethyl ammonium bromide method [46] and used as a template for the PCR amplification of *BnAHAS1*, *BnAHAS2* and *BnAHAS3* by the three pairs of primers reported by Hu et al. [29] (Appendix A). The PCR mixture (20 μL) contained 50 ng template DNA, 150 μM dNTPs, 1 × PCR buffer, 0.15 μM each primer and 0.25 U EX Taq DNA polymerase (Takara, China). PCR amplification was carried out by predenaturation at 94 °C for 5 min followed by 35 cycles at 94 °C for 30 s, 54 °C for 40 s, 72 °C for 140 s and a final extension at 72 °C for 10 min. The PCR amplification experiment was conducted in triplicate for each of the three *BnAHAS* genes. The PCR products were purified from agarose gel using the Gel Extraction Kit (TianGen, China). Subsequently, PCR products were ligated to the pMD™ 19T Vector (Takara, China) and transformed into DH5α competent cells. Five positive clones from each PCR product of each *BnAHAS* gene were selected for sequencing by the ShengGong Company (Shanghai, China). 

The obtained sequences were analyzed using the DNAMAN 6.0. The *AHAS* sequence of *A. thaliana* (*At3g48560*) was downloaded from TAIR [47]. Multiple sequence alignments were performed using the BioEdit 7.0 [48].

### 4.6. Development of CAPS Markers for BnAHAS1^544^

The sequence comparison of *BnAHASs* revealed that the nucleotide sequence 540-544 bps from the translation starting site in the WT allele of *BnAHAS1* in the line *ZS9* was GGTCC, which was mutated to GGTC*T* in the mutant line *K5* (see Section 2.5). Therefore, the restriction endonuclease *Ava*II with the restriction sites GGWCC was employed to develop the CAPS marker for the detection of the causal point mutation in *BnAHAS1*. A pair of AS-PCR primers (*BnA1*F1 and *BnA1*R4, Appendix A) was designed to amplify the target sequence containing the mutant locus. The PCR mixture (40 μL) contained 100 ng template DNA, 0.75 μM for each primer and 20 μL Primer Star Max (Takara, China). PCR amplification was carried out with 36 cycles at 98 °C for 10 s, 56 °C for 5 s and 72 °C for 10 s. All PCR products were digested with 10 U *Ava*II (New England Biolabs, Ipswich, MA, USA) for about 30 min at 37 °C at a final volume 25 μL. Subsequently, the products were separated on 2.5% agarose gel, stained with ethidium bromide and visualized using a gel imaging system (Alpha Innotech, Shanghai, China).

### 4.7. Arabidopsis Transformation and TBM Treatment

The *BnAHAS1^544T^* from the positive pMD19T-*BnAHAS1^544T^* clone was first prepared by a fusion PCR with two pairs of primers (*ahas1*NcoI-F and *ahas1*NcoI-R and *ahas1*Blunt-F and *ahas1*Blunt-R) to avoid the restriction sites of *Nco*I in *BnAHAS1* (Appendix A), which was then introduced into the expression vector pCAMBIA3301 by using the *Nco*I and *Pm*II double-digested method. The final overexpression vector was transformed into *Agrobacterium tumefaciens* GV3101. The obtained positive clones were used to introduce *BnAHAS1^544T^* into *A. thaliana* (Col-0) by using the floral dip transformation method [49]. The herbicide Basta was employed for screening positive transgenic plants. The obtained positive lines were treated with 0.3 g a.i. ha^−1^ TBM at the 4–6 leaf stage to determine whether *BnAHAS1^544T^* was the causal resistance allele to TBM. For the fresh weight assay, the obtained positive lines were treated with 0, 1.5, 3.0, 6.0 and 15.0 g a.i. ha^−1^ TBM, and WT *A. thaliana* (WT) plants were treated with 0.003, 0.03, 0.3, 0.6 and 1.5 g a.i. ha^−1^ TBM with a volume of 0.5 mL for each plant.

### 4.8. RNA Extraction and SQ-PCR

The young leaves of the homozygous *BnAHAS1^544T^* transgenic and *WT A. thaliana* plants were collected for RNA isolation by using the plant RNA extraction kit (E.Z.N.A.^®^Plant RNA Kit, R6827-01, Omega Bio-Tek, Norcross, GA, USA). The GoScript TM Reverse Transcription System (A5001, Promega, Madison, WI, USA) was utilized for the synthesis of the first-strand cDNA. The SQ-PCR primers TA1-F and TA1-R (Appendix A) were used to amplify the *BnAHAS1*. The housekeeping gene *ubiquitin-conjugating enzyme 21* (*UBC21*) was employed as the reference gene. The SQ-PCR primers, UBC_qPCR-F and UBC_qPCR-R (Appendix A), were used to amplify *UBC21*.

### 4.9. Fast-Screening Method for TBM-Resistant Rapeseed in Greenhouse

The seeds of rapeseed lines *ZS9* and *K5* and their F_1_, BC_1_ and F_2_ were prepared. Field soil and commercial-scale substrate soil (Pindstru, Ryomgaard, Denmark) were mixed at a ratio of 1:1 and subsequently filled into pots (18 × 16 cm^2^). Each pot was sown with 10–15 seeds and irrigated with a 300 mL TBM solution at different dosages. The mutant line *K5* was treated with 0, 0.03, 0.15, 0.75 and 3.75 g a.i. ha^−1^ TBM solution, whereas the line *ZS9* was treated with 0, 0.0003, 0.003, 0.03, 0.15, 0.75 and 3.75 g a.i. ha^−1^ TBM solution. Finally, the 0.15 g a.i. ha^−1^ TBM was used to test the herbicide resistance of the F_1_, BC_1_ and F_2_ derived from the parent lines *ZS9* and *K5* on the basis of the abovementioned results. All plants were cultivated in a 25 °C/10 °C (day/night) phytotron with a 16-h light/8-h dark cycle (light intensity of 4000–6000 lx).

### 4.10. Data analysis

The independent-sample *t*-test was employed to compare the *I*_50_ values of fresh and dry weights between the two lines by using the SPSS 20.0 [45]. A completely randomized design and one-way ANOVA were performed using the SPSS 20.0 software to analyze the tested traits (phytotoxicity index, the leaf angle, leaf number, fresh and dry weights) and AHAS relative activity [45], with herbicide treatment as the fixed factor and biological replication as the random factor. The means of the treatments were compared using the Tukey method.

## 5. Conclusions

The rapeseed mutant line *K5* acquired its herbicide resistance from the mutation of *BnAHAS1^544T^*, leading to the Pro–182–Ser alteration in BnAHAS1. In addition, the AS–CAPS marker was developed on the basis of the mutant allele *BnAHAS1^544T^* and the method for fast-screening TBM-resistant plants in the candidate population at the cotyledon stage, which may promote the future development of herbicide-resistant rapeseed cultivars.

## Figures and Tables

**Figure 1 ijms-21-01345-f001:**
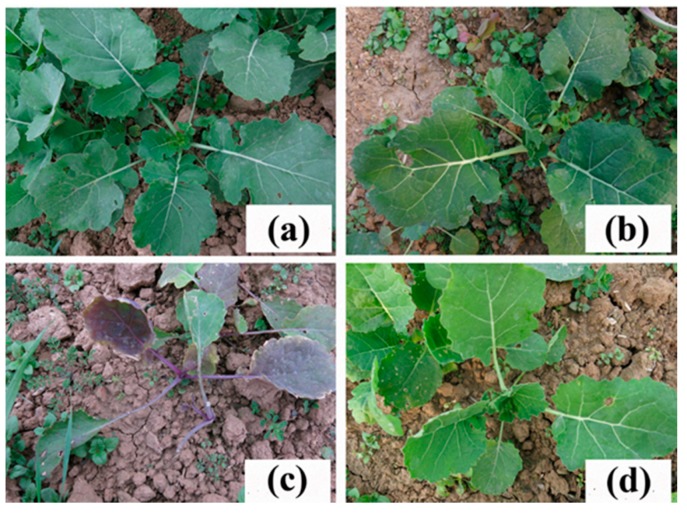
Resistance of the rapeseed mutant line *K5* to tribenuron-methyl (TBM). (**a**) Untreated wild-type *ZS9* plants. (**b**) Untreated line *K5* plants. (**c**) *ZS9* plants treated with 0.6 g a.i. ha^−1^ TBM. (**d**) *K5* plants treated with 0.6 g a.i. ha^−1^ TBM.

**Figure 2 ijms-21-01345-f002:**
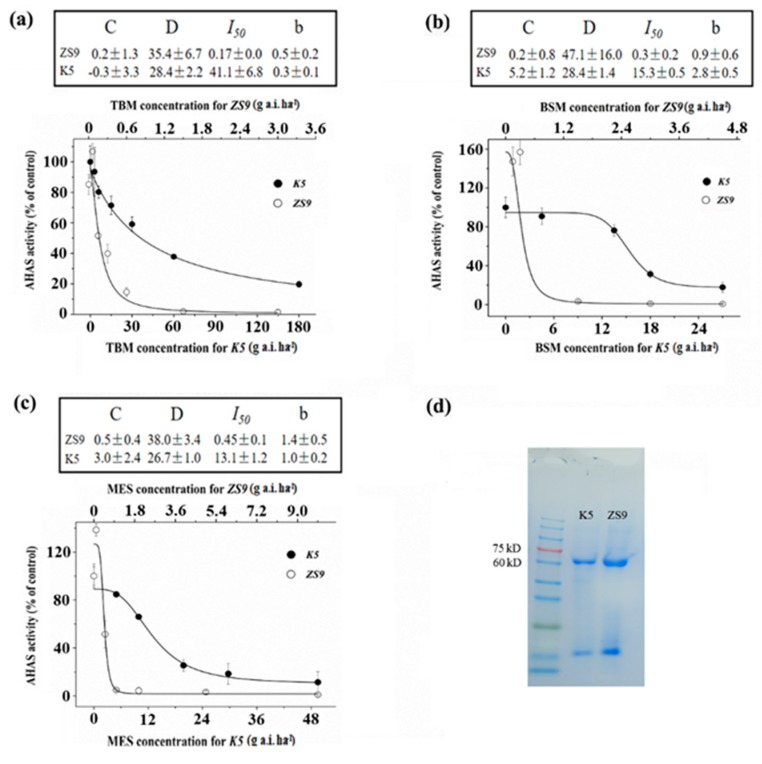
Activity assay of AHAS of *ZS9* and *K5* after spraying with three different herbicides. (**a**–**c**) Relative AHAS activity of rapeseed *ZS9* and the mutant *K5* after spraying with tribenuron-methyl (TBM) (**a**), bensufuron-methyl (BSM) (**b**) and monosulfuron-ester sodium (MES) (**c**). (**d**) Sodium dodecyl sulfate–polyacrylamide gel electrophoresis (SDS–PAGE) of AHAS enzyme extracted from fresh leaves of herbicide-free *K5* and *ZS9*. In the absence of herbicides, no significant difference was detected in the AHAS average specific activity (U/mg protein/h) between *K5* (0.075 ± 0.013, *n* = 4) and *ZS9* (0.098 ± 0.011, *n* = 4).

**Figure 3 ijms-21-01345-f003:**
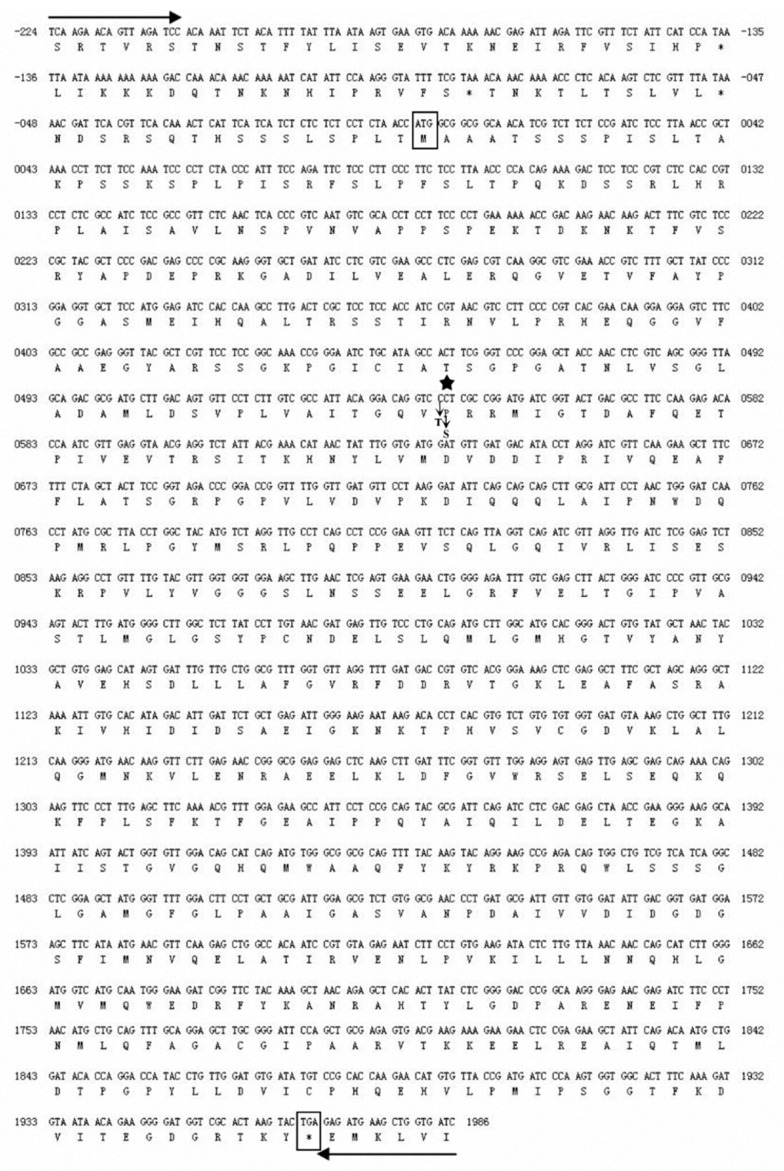
Mutant position of *BnAHAS1^544T^* in the mutant *K5*. Primer nucleotide sequences for amplifying *BnAHAS1* are indicated by the forward and reverse arrows. The boxes represent the initiation or the stop codons of the *BnAHAS1^544T^* gene. The star indicates the position (544^th^ of the *BnAHAS1^544T^* coding sequence in *K5*) of the single base substitution from C–T. The amino acid is substituted from proline (P) to serine (S).

**Figure 4 ijms-21-01345-f004:**
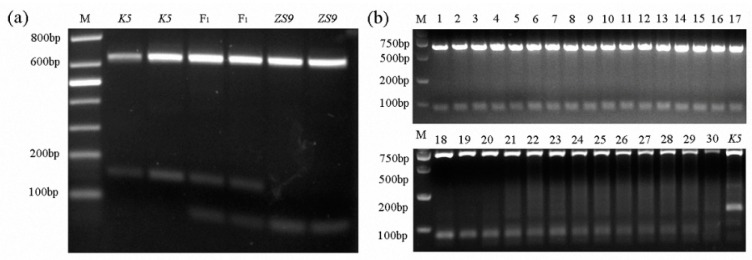
Development and verification of co-dominant AS–PCR markers. (**a**) Distinguishing susceptive and resistant parents and their F_1s_. M, marker; *ZS9*, *Zhongshuang No.9*, the susceptive line and *K5*, the mutant resistant line. (**b**) Verification of AS-PCR markers in 30 susceptive accessions. M, marker; *K5*, the mutant resistant line and lanes 1–30, 30 rapeseed susceptive accessions (Appendix A).

**Figure 5 ijms-21-01345-f005:**
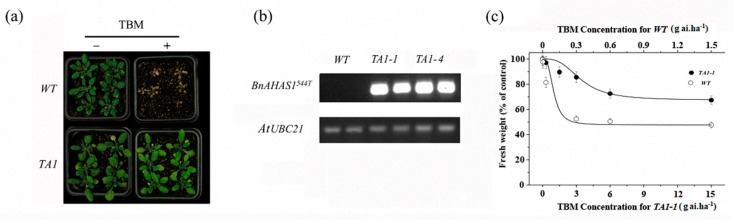
Tribenuron-methyl (TBM) resistance in transgenic *Arabidopsis* plants expressing the mutant allele *BnAHAS1^544T^*. (**a**) Phenotype of wild-type (*WT*) and transgenic *Arabidopsis* line (*TA1*) after spraying with 0.3 g a.i. ha^−1^ of TBM at the 4–6 leaf stage. (**b**) Expression analysis of the *BnAHAS1^544T^* gene in *WT* and *TA1* by semiquantitative PCR. *At**UBC21*, *ubiquitin-conjugating enzyme 21*. (**c**) Fresh weight inhibition of *WT* and *TA1-1* two weeks after spraying with different rates of TBM.

**Figure 6 ijms-21-01345-f006:**
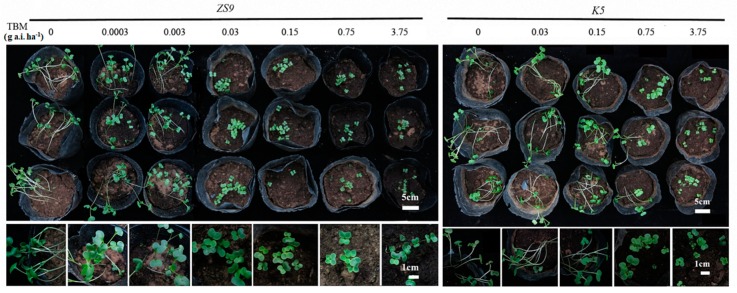
Phenotype of *ZS9* and *K5* at the cotyledon stage after irrigating with tribenuron-methyl (TBM) solution. Seeds were sown in the pots, and each pot was immediately irrigated with 300 mL TBM solution at different rates. Photographs were taken two weeks after seed sowing. *ZS9*, *Zhongshuang No.9* and *K5*, the mutant line.

**Table 1 ijms-21-01345-t001:** Cross-resistance of the mutant line *K5* to three different herbicides.

Herbicide	Lines	*I*_50_ (g a.i. ha^−1^)		Lethal Rate (g a.i. ha^−1^)	Safety Rate (g a.i. ha^−1^)
		Fresh Weight	Dry Weight		
TBM	*ZS9*	0.36 ± 0.06	0.30 ± 0.06	0.60	0.06
	*K5*	25.38 ± 11.67 **	28.98 ± 12.99 **	60.00–150.00	15.00
BSM	*ZS9*	0.27 ± 0.12	0.30 ± 0.15	1.50	0.15
	*K5*	14.91 ± 0.00 **	8.79 ± 3.12 **	>27.00	18.00
MES	*ZS9*	0.39 ± 0.12	0.36 ± 0.12	1.00	0.10
	*K5*	19.05 ± 4.59 **	19.74 ± 2.61 **	30.00–50.00	10.00

TBM, tribenuron-methyl; BSM, bensufuron-methyl; MES, monosulfuron-ester sodium; and *ZS9*, *Zhongshuang No.9*. *I*_50_ is the herbicide rate required to reduce fresh/dry weight by 50%. ** indicated a significant difference between *ZS9* and *K5* in the tested traits at a 0.01 level.

**Table 2 ijms-21-01345-t002:** Inheritance of the tribenuron-methyl resistance in the mutant line *K5.*

Cross and Population	Resistant	Sensitive	Expected Ratio	Ratio	χ^2^_c_	P
	Plants	Plants				
***K5* × *ZS9***						
F_1_	89	0	-	-	-	-
RF_1_	94	0	-	-	-	-
BC_1_	114	103	1:1	1.10:1	0.28	0.60
F_2_	1059	350	3:1	3.02:1	0.02	0.89
***K5* × *SH11***						
F_1_	172	0	-	-	-	-
RF_1_	164	0	-	-	-	-
BC_1_	177	172	1:1	1.03:1	0	1
F_2_	403	127	3:1	3.17:1	0.25	0.62

χ^2^_0.05,1_ = 3.84 and χ^2^_0.01,1_ = 6.63. *ZS9*, *Zhongshuang No*.9, *K5*, the mutant line, -, not applied and RF_1_, reciprocal F_1_.

**Table 3 ijms-21-01345-t003:** Herbicides and their rates in a volume of 300 L ha^−1^ used for a cross-resistance test for the lines *K5* and *ZS9.*

Herbicides	Rates for *ZS9*	Rates for *K5*	Herbicides	Rates for *ZS9*	Rates for *K5*
	(g a.i. ha^−1^)	(g a.i. ha^−1^)		(g a.i. ha^−1^)	(g a.i. ha^−1^)
Tribenuron-methyl	0	0	Florasulam	0	0
	0.06	3		0.6	6
	0.15	6		1.5	9
	0.3	15		3	15
	0.6	30	Sulfometuron methyl ester	0	0
	1.5	60		150	150
	3	150		300	300
Bensufuron-methyl	0	0		600	600
	0.15	4.5	Nicosulfuron	0	0
	0.3	13.5		13.5	13.5
	0.6	18		27	27
	1.5	27		54	54
	3	-		-	-
	4.5	-		-	-
Carfentrazone-ethyl	0	0	Monosulfuron-ester sodium	0	0
	1.5	1.5		0.1	5
	3	3		0.2	10
	4.5	4.5		0.5	20
Imidazole nicotinic	0	0		1	30
	49.5	49.5		2	50
	-	-		5	-
	-	-		10	-
Glyphosate	0	0	Glufosinate	0	0
	15	15		6	6
	21	21		9	9
	27	27		15	15

*ZS9*, *Zhongshuang No*.9; *K5*, the mutant line and -, no data.

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
