# Peer review of "Inheritance and Molecular Characterization of a Novel Mutated AHAS Gene Responsible for the Resistance of AHAS-Inhibiting Herbicides in Rapeseed (Brassica napus L.)"

_ijms, 2020, doi:10.3390/ijms21041345_

Round 1

Reviewer 1 Report

The present study is a typical work on herbicide resistance mechanisms. The approach is interesting and the quality adequate. Authors should revise the manuscript for some English-language mistakes and also some expression's mistakes (such as the "use of herbicides" or "chemical control" instead of "utilization of herbicides"). I would also like to see a stronger discussion section and talking about the other mechanisms of resistance that are potentially involved. I suggest the acceptance of the paper after its minor revision.

Author Response

Response to Reviewer 1 Comments

International Journal of Molecular Sciences / ijms-699460

Inheritance and molecular characterization of a novel mutated AHAS gene responsible for the resistance of AHAS-inhibiting herbicides in rapeseed (Brassica napus L.)

Open Review 01:

Comments and Suggestions for Authors

The present study is a typical work on herbicide resistance mechanisms. The approach is interesting and the quality adequate. Authors should revise the manuscript for some English-language mistakes and also some expression's mistakes (such as the "use of herbicides" or "chemical control" instead of "utilization of herbicides"). I would also like to see a stronger discussion section and talking about the other mechanisms of resistance that are potentially involved. I suggest the acceptance of the paper after its minor revision.

Response:

Thank you very much for your positive comments and your good suggestions. We have thoroughly revised the manuscript to the best of our ability. We have corrected some English-language mistakes in the revised manuscript according to your suggestions, such as “use of herbicides” instead of “utilization of herbicides”.

We have added a paragraph to discuss about the other mechanisms of resistance that are potentially involved, detailed in “Discussion section”, lines 647-681 in the revised version. We have added four references to support the discussion.

Christophe Delye (2013) Unravelling the genetic bases of non-target-site-based resistance (NTSR) to herbicides: a major challenge for weed science in the forthcoming decade. Pest Manag Sci 69: 176–187 Christophe Delye, Marie Jasieniuk, and Valerie Le Corre(2013) Deciphering the evolution of herbicide resistance in weeds. Trends in Genetics,2013,29(11):649-658 Brent P. Murphy and Patrick J. Tranel (2019) Target-Site Mutations Conferring Herbicide Resistance. Plants,8,382 Mithila Jugulam and Chandrima Shyam (2019) Non-Target-Site Resistance to Herbicides:Recent Developments. Plants,8,417

Reviewer 2 Report

-The article was much improved compared to the previous version. And I want to congratulate the authors for that. However, I want the authors to be aware that it should be easier to read and not mess up the readers. Ideas must be clear and straightforward as well as phrases. Avoiding repetitions that make the reader confuse.

-Other point is that the authors seem not to have done certain trials. An example is when they designate one of the herbicides as imidazoline (Table 1). Imidazolinone is a group of AHAS-Inhibiting herbicides. 

-Review that there can be no comma before "and" when it comes to enumerations

-Homogenize size and type font

-Review that the herbicide names are correctly written

Others:

INTRODUCTION

-Line 41: Change "chlorpyridine acid and benazolibn-ethyl" by "chloropyridine acid and benazolin-ethyl"

-Line 50: Delete "kind" in "this kind of herbicides"

-Line 59: Delete "this kind of herbicides" at the end.

-Line 64-65: Delete "suggesting the enhancement of plant herbicide- resistance after spraying with herbicides" because is understood

RESULTS

-Line 97: Delete "kinds" because they are the same kind of herbicides (sulfonylureas)

-Line 98: Change "monosulfruon-sodium" by "monosulfuron-sodium"

-Line 100: Please use the same type and size font

-Line 101: Try to change "Three weeks" by "21 DAT (days after treatment)"

Table 1: This table must appear in Materials and Method. The authors should be more attentive to the Table format and to use bold correctly, in addition to the names of the herbicides (Change "Monosulfon" by "Monosulfuron" ). What imidazolinone was used?

-Line 110-112: "The analysis of variance indicated that remarkable differences existed among the tested rates of three different herbicides for lines ZS9 and K5 except in leaf angle, leaf numbers, fresh weight, and dry weight of K5 under BSM treatments (Tables S1 and S2)."

This phrase is not clear. Are there differences in the phytotoxiciy index between ZS9 and K5 and no in the others traits?

MATERIALS AND METHODS

-Line 396: What "were the same across trials"? (type of soil , how is the preparation, fertilizer??, and irrigation).

Author Response

Response to Reviewer 2 Comments

International Journal of Molecular Sciences / ijms-699460

Inheritance and molecular characterization of a novel mutated AHAS gene responsible for the resistance of AHAS-inhibiting herbicides in rapeseed (Brassica napus L.)

Open Review 02:

Point 1

Comments and Suggestions for Authors

-The article was much improved compared to the previous version. And I want to congratulate the authors for that. However, I want the authors to be aware that it should be easier to read and not mess up the readers. Ideas must be clear and straightforward as well as phrases. Avoiding repetitions that make the reader confuse.

Response: Thank you very much for your positive comments and your good suggestions. We have thoroughly revised the manuscript to the best of our ability. We hope that the revised version of our manuscript will meet the requirements for publication in the journal.

Point 2

-Other point is that the authors seem not to have done certain trials. An example is when they designate one of the herbicides as imidazoline (Table 1). Imidazolinone is a group of AHAS-Inhibiting herbicides. 

Response:

Thank you very much for your comments. In the research, we used imidazole nicotinic, which is one of Imidazolinone herbicides. We have revised this in the version of manuscript (the revised Table 3).

Point 3

-Review that there can be no comma before "and" when it comes to enumerations.

Response:

Thank you! We have checked our manuscript and corrected them in the revised version of the manuscript.

Point 4

-Homogenize size and type font

Response:

We have homogenize size and type font in the in the revised version of the manuscript.

Point 5

-Review that the herbicide names are correctly written.

 Response:

Thank you very much for your patience! We have checked our manuscript and corrected them in the revised version of the manuscript.

Point 6 Others:

 INTRODUCTION

 -Line 41: Change "chlorpyridine acid and benazolibn-ethyl" by "chloropyridine acid and benazolin-ethyl"

 Response:

Thank you very much! We have corrected them. And we have checked our manuscript and corrected them in the revised version of the manuscript.

-Line 50: Delete "kind" in "this kind of herbicides". -Line 59: Delete "this kind of herbicides" at the end.

 Response:

Here, line 51, “this kind of herbicides” refers to acetohydroxyacid synthase -inhibiting herbicides. If we delete "kind" in "this kind of herbicides", which will make the readers confuse. So, we keep it in the revised version.

-Line 64-65: Delete "suggesting the enhancement of plant herbicide- resistance after spraying with herbicides" because is understood.

 Response:

Thanks for your suggestion! Lines 67-68, we deleted it in the revised version.

RESULTS

-Line 97: Delete "kinds" because they are the same kind of herbicides (sulfonylureas)

 Response:

 Yes. Line 102, we deleted it.

-Line 98: Change "monosulfruon-sodium" by "monosulfuron-sodium"

 Response:

 Thank you very much! We have checked our manuscript and corrected them in the revised version of the manuscript.

-Line 100: Please use the same type and size font

 Response:

We have checked our manuscript and homogenized size and type font in the manuscript.

-Line 101: Try to change "Three weeks" by "21 DAT (days after treatment)"

 Response:

Thank you for your suggestion. We changed “three weeks spraying with herbicides” by "21 DAT (days after treatment)" in the revised version of manuscript.

Table 1: This table must appear in Materials and Method. The authors should be more attentive to the Table format and to use bold correctly, in addition to the names of the herbicides (Change "Monosulfon" by "Monosulfuron" ). What imidazolinone was used?

 Response:

Thank you for your suggestion. We have moved this table to the “Materials and methods” section, lines 437-439. Its name is changed to Table 3. Table format was also corrected. “Monosulfon” was changed to “Monosulfuron”. In the research, we used imidazole nicotinic, which belongs to Imidazolinone herbicide.

-Line 110-112: "The analysis of variance indicated that remarkable differences existed among the tested rates of three different herbicides for lines ZS9 and K5 except in leaf angle, leaf numbers, fresh weight, and dry weight of K5 under BSM treatments (Tables S1 and S2)."

This phrase is not clear. Are there differences in the phytotoxiciy index between ZS9 and K5 and no in the others traits?

Response:

 We have modified this paragraph. “Results” section, lines 113-117. Sentence was modified as “The analysis of variance indicated that only BSM treatments have no effect on leaf angle, leaf numbers, fresh weight and dry weight of K5, except this, traits of lines ZS9 and K5 under other tested rates of herbicides were remarkably affected (Tables S1 and S2)”.

MATERIALS AND METHODS

-Line 396: What "were the same across trials"? (type of soil , how is the preparation, fertilizer?, and irrigation).

Response:

Thank you for your suggestion. Section “Materials and methods”, lines 741-745, we have modified this sentence as “Before sowing, 600 kg ha−1 of N, P and K compound fertilizer (N:P:K = 15:15:15) and 75 kg ha−1 urea with nitrogen content of 46.7% were applied as a basal fertilizer. At the end of December, rapeseed seedlings were watered for their overwintering.”

Round 2

Reviewer 2 Report

I want to inform  to the authors that now the article is conditioned to my requests and therefore I have no other objection to it. For my part it is can be accepted and I would like to congratulate the authors for this work in which they have had to change their customs a little to those of agronomy area. Congratulations.

This manuscript is a resubmission of an earlier submission. The following is a list of the peer review reports and author responses from that submission.

Round 1

Reviewer 1 Report

The topic is clearly interesting, however the overall quality is very poor and the paper really chaotic in several points. Before any other criticism, the paper requires an extensive English-language revision. Only the abstract is relatively sufficient. In many cases throughout the text, there are serious mistakes (some indicative comments below). Some more references can be also added. Under my point of view, authors are encouraged to build their manuscript from the beginning and put considerably more effort.

Indicative mistakes

Comment 1: susceptive to all application rates of the tested herbicides (Table S2): probably meaning susceptible

Comment 2: about three weeks after sprayed with herbicides: there is no "about" in weed science

Comment 3: individual plants were observed no apparent difference with TMBfree ones even the treated TBM concentration was up...: chaotic, plants were not different maybe?

Comment 4: talking about herbicides, concentrations and rates are usually expressed as g a.i./ha and not per volume

Comment 5: l.111: it was obviously that the inhibition rate: obvious?

Comment 6: Figure 1: resistance performance??? This is the efficacy of the herbicides, resistance has no...performance!

Comment 7: l.125: The line K5 also showed a certain resistance: hos is certain resistance defined????? Never heart of that

Comment 8: l.136: Generally, the mutant line K5 conferred a relatively high resistance to TBM and a certain resistance: again as above

Comment 9: l.138: approached the recommended dosage for weeds control in wheat field: for weed control in wheat crop

Comment 10: l.232: Nevertheless, it is largely unknown that whether the herbicide-resistance is determined by the mutation: largely unknown or widely???? whether, not "that whether"

Comment 11: l.265: It is a time- and labor-consuming works to find natural or mutated herbicide-resistant individuals in field by the traditional way: you never start a results' section that way and which is the traditional way???

Comment 12: l.270: all treated seeds began emergence: started to emerge etc

Reviewer 2 Report

The manuscript is very interesting and very well designed, although certain aspects must improved so that readers do not get lost and know how the study is done in a correct way.

-Firstly the authors must homogenize the typeface look at lines 66 (In B. napus ....) - 72. The same in line 101 (NSF, glyphosate,...).416-419, ... Please review it.

-Review the English grammar please.

-One point very important for me is the used method in the point 4.2 and 4.3. The authors spray the plants with the herbicides, but they do not comment about the volume of application of the herbicides, the dose per area,...I recommend to the authors that a most appropriate explanation of this experiment be used and also the units for the foliar application of herbicides. For example:  20 mg a.i. ha-1 in a volume of 200 L ha-1. Please change it and correct the units. 

-A comparison between the normal field dose and the used dose would be useful to see the degree of resistance in that mutant

-In my opinion the Table S1 would be a principal Table, not a supplementary.

I insist that this manuscript is very interesting and I encourage the authors to modify some points and submit it again.